# A Method for Intelligent Road Network Selection Based on Graph Neural Network

**Xuan Guo** [1,2] ⑩, **Junnan Liu** [2,3,]* ⑩, **Fang Wu** [3] **and Haizhong Qian** [4]

1   Institute of Computer and Artificial Intelligence, Zhengzhou University, Zhengzhou 450001, China
2   State Key Laboratory of Geo-Information Engineering, Xi'an 710054, China
3   Institute of Earth Science and Technology, Zhengzhou University, Zhengzhou 450001, China
4   Institute of Geospatial Information, Information Engineering University, Zhengzhou 450001, China
\*   Correspondence: ljnzzu@zzu.edu.cn; Tel.: +86-0371-6368-5366

**Abstract:** As an essential role in cartographic generalization, road network selection produces basic geographic information across map scales. However, the previous selection methods could not simultaneously consider both attribute characteristics and spatial structure. In light of this, an intelligent road network selection method based on a graph neural network (GNN) is proposed in this paper. Firstly, the selection case is designed to construct a sample library. Secondly, some neighbor sampling and aggregation rules are developed to update road features. Then, a GNN-based selection model is designed to calculate classification labels, thus completing road network selection. Finally, a few comparative analyses with different selection methods are conducted, verifying that most of the accuracy values of the GNN model are stable over 90%. The experiments indicate that this method could aggregate stroke nodes and their neighbors together to synchronously preserve semantic, geometric, and topological features of road strokes, and the selection result is closer to the reference map. Therefore, this paper could bridge the distance between deep learning and cartographic generalization, thus facilitating a more intelligent road network selection method.

**Keywords:** cartographic generalization; road network selection; graph neural network; deep learning

## 1. Introduction

A map is a generalized, simplified abstraction of reality that aims to reduce the realistic details [1]. Cartographic generalization, producing maps from larger to smaller scales, is one of the central concepts in map design. It aims to preserve the characteristics and patterns during the scaling process; thus, it is subjective as well as scale related. The main difficulty of cartographic generalization mainly reflects on the subjective, flexible, and fuzzy thinking activities of human beings. Moreover, multi-scale spatial representation always involves a transformation of spatial data from one scale to another. To date, this field has been widely studied from various aspects of geographic datasets (such as selection, simplification, and merging). As essential parts of a map, roads are widely considered coherent networks [2]. An appropriate level of road network detail could facilitate transportation-related applications. Compared with simplification and merging, road network selection is the first and most crucial step, reducing road elements by deleting relatively unimportant parts [3]. We would give up the roads with low class, short length, poor connectivity, and weak continuity, which seem to be relatively unimportant objects.

During road network selection, a structure with the property of 'good continuity' (referred to as stroke) is introduced to combine several continuous and short segments, which are simultaneously deleted or selected by geometric or semantic stroke features [4]. The multiple features of stroke are also used to judge its selection labels, thus improving the generalization results [5]. However, stroke features cannot reflect all aspects of the network structure, thus posing several challenges to the road network selection. There are two major challenges for road network selection to be settled urgently. The one challenge is that the

road neighborhood varies significantly in network structure, ignoring close relationships between road strokes. The other challenge is that the existing intelligent selection models ignore topological relationships and are hardly applied to the road network in noneuclidean space. Therefore, it is urgent to shift from the traditional to a graphic structure, making selection results more in line with the actual road network situation.

Deep learning is an essential embodiment of modern artificial intelligence, which realizes the automatic learning of classification tasks through neural networks [6,7]. It could be widely used in geoscience and engineering, such as pavement designing [8], landslide modeling [9], and traffic accident risk forecasting [10]. As a binary classification task, we could implement intelligent road network selection by neural networks. Graph neural network (GNN), an emerging deep learning technology, is suitable for network structures (including road networks and their dual graph) and could simultaneously evaluate semantic, geometric, and topological features, thus shifting selection from single-type to multiple-type features. Among them, the class represents the semantic feature, and the higher the class, the more important the stroke would be. Length represents the geometric feature, and the longer the stroke length, the greater the selection possibility. Degree represents the topological feature, and the higher the connectivity, the more likely to be selected. Motivated by these factors, an improved selection method for generalizing road networks based on GNN is proposed. Firstly, the road network selection case is designed based on extracted strokes, and a sample library for GNN is constructed from stroke cases. Then, in order to solve the problem of road neighbors varying significantly, neighbor sampling and aggregation rules are developed to automatically update stroke features based on attribute characteristics and spatial structure. Finally, a road network selection deep learning model is designed, which includes multiple GNN layers, a fully connected neural network classification layer, and a normalization layer. This GNN-based model could calculate the selection results of each stroke to realize intelligent road network selection and make results more aligned with the actual situation. The specific contributions are the following:

- A road network selection deep learning framework is designed, which could become a reference for other tasks involved in network structure.
- A sampling method is proposed in the road network, by which relationships between strokes could be established.
- A GNN-based road network selection model is designed, considering both attribute characteristics and spatial structure of the vector road network.

The rest of this paper is organized as follows. Section 2 briefly surveys the related work on road network selection and intelligent cartographic generalization. The GNN-based road network selection method is proposed in Section 3, which consists of constructing a sample library for cases, designing neighbor sampling and aggregation rules, and establishing a road network selection deep learning model. Section 4 provides comparative analyses and discusses the detailed results, while Section 5 concludes the paper.

## 2. Related Work

Cartographic generalization has always drawn attention from researchers, in which road network selection is a crucial step. This section reviews several research directions and related backgrounds.

### 2.1. Road Network Selection

In road network selection, several researchers have widely explored ways of deleting or selecting roads to preserve the shape characteristic and pattern of map elements. According to the road representation mode, selection methods can be broadly categorized into stoke-based, mesh-based, and integrated stroke–mesh methods. The stroke-based method is based on the principle of 'Good Continuity' to meet the cognition criteria of the Gestalt principle. Thomson et al. [4] introduced stroke, a continuous and long road composed of several segments, into the road selection. Instead of individual road seg-

ments, stroke has become the most commonly used in road network selection, by which the segments are treated as a unit to delete or select simultaneously. Hence, the continuity characteristics of roads can be preserved after generalization. The mesh-based method models the road network as a set of meshes, which are indirectly determined by aggregating meshes with specific constraints (e.g., mesh density is higher than the pre-defined threshold). By representing the geometric structure as meshes, Chen et al. [11] relatively merged high-density meshes to implement the road network selection process. However, the stroke-based method performs better for linear patterns, and the mesh-based method works for areal patterns. To preserve linear and areal patterns, the integrated stroke–mesh method, proposed by Zhou et al. [12], handles linear hierarchy with stroke-based and areal segments with mesh-based. This integrated strategy could combine stroke-based and mesh-based methods in an integrated concept, thus performing better than the individual for road network generalization [13].

Under the foundation of sufficiently utilizing road network characteristics, the selection process focuses on semantic and geometric characteristics, thus improving the generalization results [5]. To consider topology, Mackaness and Beard [14] represented a road network based on graph theory, in which road segments and intersections are modeled as edges and nodes, respectively. By contrast, Jiang and Claramunt [15,16] designed a dual graph structure to quantify road importance, representing roads and intersections as nodes and edges, respectively. Moreover, the dual graph structure substitutes strokes for individual road segments as nodes or edges to preserve continuity and connectivity. Additionally, with the development of volunteered geographic information, big data such as traffic flow, taxi trajectory, and points of interest have been adapted as indicators to widen road network selection [17–19]. Hence, multiple criteria decision-making methods are introduced to regulate the influence of different indicators, such as information entropy, coefficient of variation, and analytic hierarchy process.

The above-proposed methods focus mainly on semantic, geometric, and topological characteristics. However, the characteristics could only be evaluated in the attribute dimension. On the one hand, the road network has significant spatial structure characteristics, which should also be evaluated. Therefore, we need to assess the importance of stroke, taking both attribute and spatial structure into consideration. On the other hand, the relationship between the road network and its dual graph is complicated. Hence, we need to ratiocinate according to the variable relationship between strokes.

### 2.2. Intelligent Cartographic Generalization

The difficulty and complexity of intelligent cartographic generalization are mainly manifested in the high dependence on human thinking activities. Although researchers have made breakthroughs in artificial intelligence since the 21st century, there is still a significant gap in the research of intelligent cartographic generalization, which will become an essential task in the future. As a primary research direction, road network selection has achieved rich results based on machine learning. For example, Deng et al. [20] proposed a road network generalization model based on a genetic algorithm, which reflects the spatial distribution characteristic and practical economic value of the original road network. By using case-based reasoning learning, Guo et al. [21,22] proposed intelligent road network selection methods based on case analogical reasoning and inductive reasoning models. These models formalize the expert experience and make the road network selection more intelligent. Zhou et al. [23,24] summarized nine supervised classification algorithms and two neural networks in road network selection, demonstrating that the selection results of the ID3 (iterative dichotomiser 3) decision tree algorithm and neural network are more stable. By learning and training samples, Liu [25] established an automatic road network selection method based on a Kernel-based machine learning classification algorithm.

With the advancements in computing power and big data, machine learning technology has developed toward deep learning with deeper levels, higher efficiency, and more robust learning ability [26]. The deep learning module can learn results from complex

structures without too much manual intervention, in which the abilities in some areas are almost comparable to humans. It is well known that different neural networks are suitable for processing different data types. For example, recurrent neural networks process text data with regular sequence structure, while convolutional neural networks manipulate image data with two-dimensional raster structures. And for road networks, several kinds of research dealt with neural networks to realize intelligent generalization. From a progressive selection perspective, Yuan et al. [27] proposed a selection model based on a radial basis function neural network for small-scale data. At the same time, Li [28] simulated the generalization process by backpropagation neural network to maintain the overall structure and realize the progressive selection of road network. From a structure characteristics perspective, Yang [29] proposed a road selection method based on SOM (self-organizing feature maps) neural network and clustering analysis. Zhang et al. [30] offered a selection strategy by graph convolutional network to use spatial characteristics, thus obtaining ideal results.

The above intelligent cartographic generalization methods have obtained specific achievements. Nevertheless, it is difficult to synchronously account for the attribute characteristics and spatial structure of road networks. Hence, it is urgent to process non-Euclidian spatial data and shift from single-type to multiple-type features, thus making road selection results more aligned with the actual road network situation.

## 3. Methods

As a hotspot in artificial intelligence, GNN applies well to road networks and dual graphs. Hence, it is introduced to improve the efficiency and accuracy of road network selection.

### 3.1. Road Network Selection Framework

Road network selection is highly uncertain and complex due to the influence of spatial cognition and visual psychology. Several differences exist in selection results between traditional methods and manual judgment. Hence, deep learning models are introduced to make selection more intelligent. This paper adopts GNN for road network selection to learn graph structure features, which could simultaneously consider both attribute characteristics and spatial structure. As shown in Figure 1, a GNN-based road network selection framework is proposed, including sample library construction, stroke features aggregation, and selection model design.

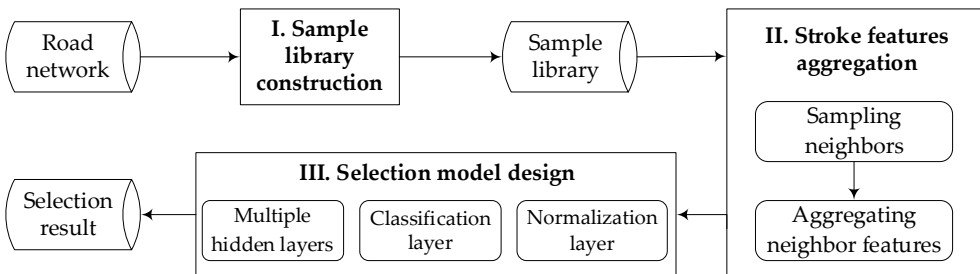

**Figure 1.** GNN-based road network selection framework.

1.  Sample library construction. The road network selection case is designed considering semantic, geometric, and topological features based on the dual graph of road stroke. Afterward, maximum –minimum normalization and one-hot coding methods are used to process case features. And a sample library for learning and training the GNN model is constructed based on these cases.
2.  Stroke features aggregation. In a dual graph, neighbor nodes are randomly sampled to obtain the strokes associated with the target node based on the road network. Then, the neighbor features are aggregated to the target stroke by rules, thus updating the target stroke feature. Therefore, this method could consider attributes and spatial structure in the road network and shift selection from single-type to multiple-type features.

3. Selection model design. The GNN-based road network selection model consists of multiple hidden GNN layers, a fully connected network classification layer, and a normalized exponential layer. The model is trained by backpropagation and optimized by cross-entropy loss function and adaptive moment estimation algorithm, thus selecting road stroke automatically.

### 3.2. Road Network Selection Sample Construction

### 3.2.1. Case Extraction

Case-based reasoning, significant research in artificial intelligence, could solve new problems by matching previous successful solutions with similar issues. Riesbeck [31] believed that the case was a set of features describing the problem state, solution process, and solution result, which could effectively solve the bottleneck of cartographic generalization knowledge acquisition. Hence, the case of road network selection (Case) is designed to express generalization knowledge. The case consists of the problem situation, solution, and solution result descriptions, represented by a triple, including selection case object (*O*), case feature (*F*), and generalization label (*L*).

$$Case : \langle O, F, L \rangle \tag{1}$$

The case takes stroke as the selection case object and selection or deletion as the generalization label, thus recording attributes and generalization results. In addition to the selection case object and generalization label, the case feature is extracted from the literature regarding road network selection to express characteristics well. Based on a statistical survey, features (including class, length, and degree) are generally adopted by most scholars to describe stroke attributes quantitatively (Table 1). Among them, class represents the semantic property, which could reflect the importance of the stroke. And the higher the level, the more important the stroke. Length represents the geometric property, which could reflect the influence range of the stroke. The longer the sum of the length of each section, the greater the influence range of the stroke. Degree represents the topological property, which is calculated by the road network dual graph and could reflect the accessibility of the stroke. The stronger the accessibility, the greater the importance of the stroke. In addition, we also take the number of neighboring residents (short for resident number) as a feature to ensure that the selection result is suitable for residents. In summary, class belongs to the semantic feature, while the length, degree, and resident number belong to numerical features.

**Table 1.** Road network selection features in the literature.

| Literature | Features |
|:---:|:---|
| [2] | class, length |
| [5] | length, travel time |
| [11] | class, length, degree |
| [15] | degree, closeness, betweenness |
| [16] | class, length, degree, closeness, betweenness, lanes, speed |
| [32] | class, length, sinuosity |
| [33] | class, length, degree, number of lanes, number of traffic directions, width |
| [34] | length, number of connections, attributes |
| [35] | betweenness |
| [36] | length, degree, closeness, betweenness, density, traffic estimation |
| [37] | class, length, Voronoi-based density |
| [38] | class, length, degree, closeness, betweenness |

To build road network selection cases, a matching method is proposed to extract same-name entities from series maps, which exist in cartographic generalization knowledge. As shown in Figure 2, we could automatically obtain cases by comparing road networks of different scales in the same area. The specific steps are as follows:

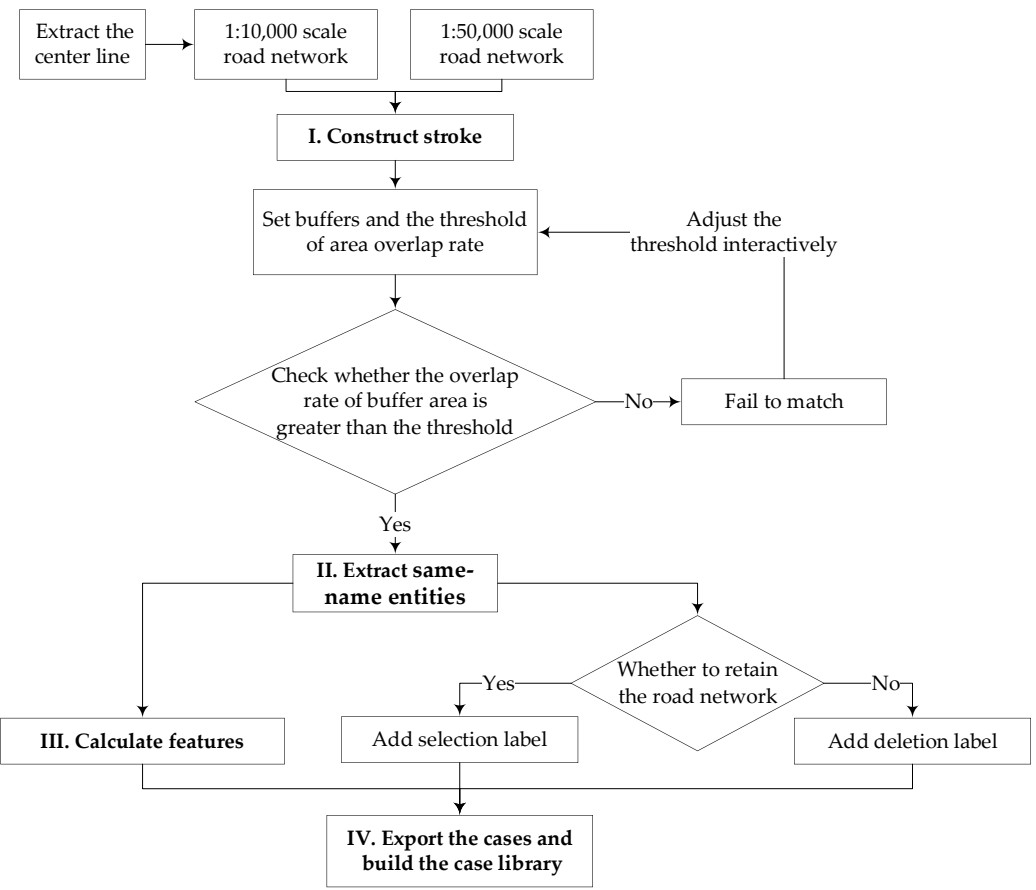

**Figure 2.** Case extraction flow of road network selection.

1.  Stroke construction. Road networks of different scales are reprocessed to construct strokes according to the 'maximum fit per pair' policy.
2.  Same-name entities extraction. According to expert experience, buffers for the road network are constructed to calculate the overlap area rate from different scales. At the same time, the threshold of overlap area rate is set as 80% based on the experiences and existing studies. When the actual calculated rate exceeds the threshold, the entity with the same name is successfully matched and given matching identifiers.
3.  Features and label calculation. Step II is repeated to traverse all strokes on a large-scale map, and their features are calculated. If the stroke has a matching identifier, a selection label is added; otherwise, it is unselected.
4.  Cases exportation and the library building. An ID is added to each stroke, and cases are exported according to Formula (1) to build a sample library.

As shown in Figure 3, the matching results on a 1:10,000 scale are demonstrated, in which selection labels to the red strokes are added, and deletion labels are added to the remaining black strokes. Therefore, we could provide road network selection cases for constructing a sample library. Generally, traditional cartographic generalization is mainly realized through operators such as aggregation, classification, enhancement, and exaggeration. Thereinto, aggregation is used to transform multiple features into one feature, while classification is used to organize selected features into categories. Enhancement is defined to clarify or elevate the symbol message, while exaggeration is defined to emphasize or maintain the characteristic aspect. These operators are used to define various methods or operations of geometric transformations. However, as a part of pre-processing, selection is usually the first step in cartographic generalization. Hence, we only consider road network selection operation between different map scales. In the process of cartographic generalization, there are some priority rules for road network selection. Firstly, we need

to select important roads, and then we need to maintain the plane graphic feature of the network. In addition, the selection should be adapted to residents. Lastly, we should maintain the road density comparison of different areas.

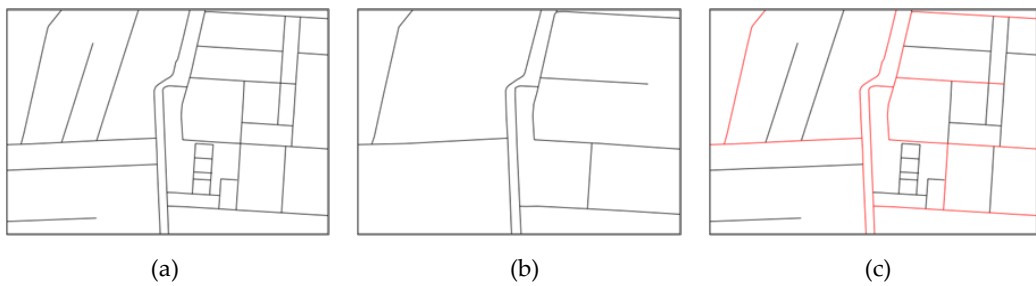

**Figure 3.** Example of road network matching results demonstrating (**a**) 1:10,000 scale road network, (**b**) 1:50,000 scale road network, and (**c**) the matching result.

### 3.2.2. Sample Construction

As a computational model, GNN could not directly calculate the road network and its features and could only deal with numerical data. Hence, it is necessary to convert cases into sample data that can be quantified and expressed in the GNN model. In road network selection, a dual graph is a frequently used road network model described by graph theory, formally defined as $G = (V, E, A)$. As shown in Figure 4, strokes and intersections are, respectively, mapped as nodes ($V = \{v_1, v_2, \cdots v_n\}$) and edges ($E = \{v_i v_j | v_i, v_j \in V\}$) in a dual graph. The edges describe and quantify relationships between strokes, making it easier to analyze road network structures. A dual graph adjacency matrix $A \in R^{n*n}$ is built for road networks from edges, where rows and columns represent nodes, and $n$ is the number of nodes. In the adjacency matrix, each element value is defined as 1 or 0, indicating whether there is a topological relationship between strokes.

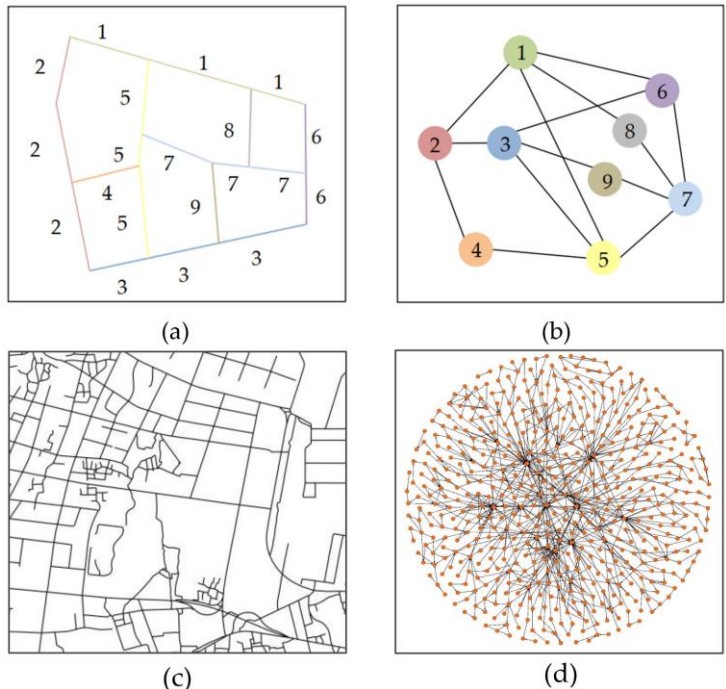

**Figure 4.** Road network example: (**a**) represents an abstract road network; (**b**) illustrates its dual graph; (**c**) represents an actual road network; (**d**) illustrates its spatial structure.

To construct a sample library, it is necessary to quantify features and generalization labels. Semantic and numerical features have different dimensions and units, leading to the long training time and model convergence failure. Hence, it is necessary to standardize

these features to eliminate their differences. The max–min standardization method maps the numerical features (including length, degree, and resident number) to [0,1]. At the same time, one-hot encoding is adopted to process class features (including national roads, provincial roads, county roads, major urban roads, and minor urban roads). In this paper, a 5-bit one-hot encoding is designed to encode road classes (e.g., [1,0,0,0,0], [0,1,0,0,0], [0,0,1,0,0], [0,0,0,1,0], and [0,0,0,0,1]). At any time, only one has a separate register bit, so the discrete and disorder features are mapped to integer values.

In addition to features, there is a great need to quantify cartographic generalization labels. A vector group $\langle x_i, y_i \rangle$ is designed to represent stroke features and selection results, where $x_i$ represents an eight bits feature vector and $y_i$ represents a two bits label vector. In detail, the feature vector could reflect the character of a stroke node, where the first three bits represent the normalized values of length, degree, and resident number, respectively. At the same time, the last 5 bits represent the one-hot encoding of the road class. Moreover, the label vector describes the generalization result, where [1,0] and [0,1] indicate selection and deletion, respectively. Then, the samples are combined from extracted features and generalization labels. Figure 5 illustrates the specific encoding of selected (stroke1) and deleted (stroke2) samples.

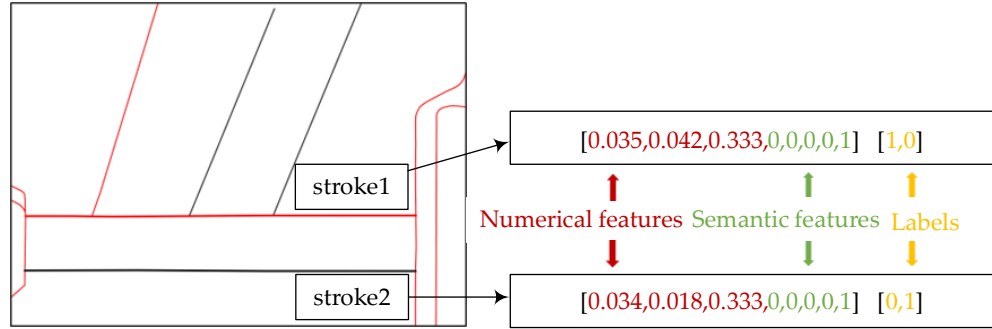

**Figure 5.** The specific encoding of constructed samples.

### 3.3. Stroke Features Aggregation

GNN mainly includes spectral and spatial convolution operations [39]. The spectral realizes convolution operation by Fourier forward inverse transform, bringing about high computational complexity and high resource utilization. The spatial has a simple calculation process and only needs to learn a few features to realize convolution operation. In addition, it is usually impossible to train our neural network model with complete data directly because the entire road network is complex, numerous, and diverse. Therefore, the spatial operation is adopted to aggregate the attribute characteristics from spatial structure, which simultaneously considers semantic, geometric, and topologic features. The adopted operation includes sampling neighbors and feature aggregation, comprehensively evaluating the stroke importances.

### 3.3.1. Sampling Neighbors

There are two methods for obtaining stroke nodes associated with the target node in the dual graph: full neighbor sampling and random sampling, where the associated stroke nodes are regarded as a layer. Full neighbor sampling starts from the target node and obtains all neighbor nodes. The sampling depth of the target node is designed as $k$, and the depth of each exterior layer is reduced by 1 in the sampling process. This way, all the affected nodes are obtained from inner to outer step by step. Figure 6a demonstrates a full neighbor sampling process with two layers. To acquire the characteristics of the target node $x_{15}^2$ in layer 2, all the nodes in layer 1 directly related to the target node are sampled, i.e., $x_{10}^1, x_{11}^1, x_{12}^1, x_{13}^1, x_{14}^1$. Then, all the other neighbor nodes in layer 0 (including $x_1^0, x_2^0, x_3^0, x_4^0, x_5^0, x_6^0, x_7^0, x_8^0, x_9^0$) are obtained in layer 1.

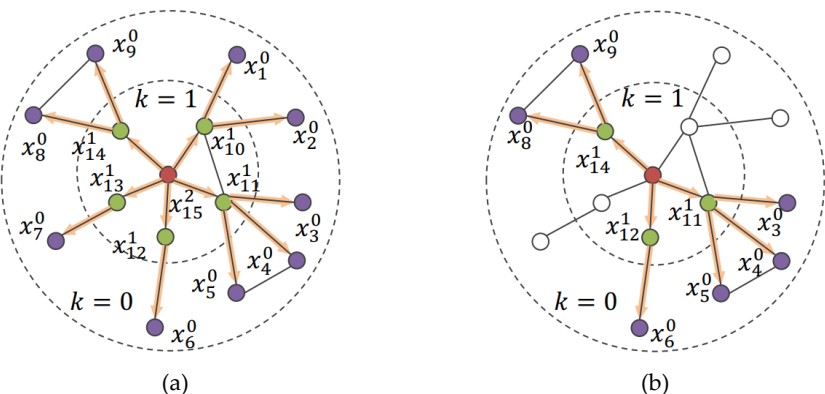

**Figure 6.** Different neighbor sampling methods: (**a**) represents complete sampling, and (**b**) illustrates random sampling.

Compared with full neighbor sampling, random sampling only acquires a part of the neighbor nodes of the target, which could significantly improve GNN training efficiency. The dual graph of the road network belongs to the undirected graph, and there is no sequence relation between stroke nodes. Hence, random sampling assumes the number of samples as $n$ and accomplishes a random process by sampling some adjacent nodes with equal probability. Thereinto, the sampling method is without replacement if there are more neighbor nodes than $n$; otherwise, it is with replacement. Firstly, the target node is taken as the center, and some neighbor nodes are obtained from the exterior. Then, subgraphs with the target node are built by randomly sampling from each node outward. Specifically, the whole dual graph is taken as an input to acquire the set of subgraphs as an output. Each subgraph only contains part nodes of two layers, which could obtain neighbor characteristics and save the computing cost. Figure 6b illustrates the number of samples (i.e., three) and the target node $x_{15}^2$. In layer 1, three nodes ($x_{11}^1$, $x_{12}^1$, $x_{14}^1$), directly adjacent to the target node, are acquired to build a sampling subgraph $G_{sub}^1$. In layer 0, three nodes ($x_3^0$, $x_4^0$, $x_5^0$) adjoined by node $x_{11}^1$ are acquired, and the sampling with replacement is conducted to gain three nodes ($x_6^0$, $x_8^0$, $x_9^0$) adjacent to $x_{12}^1$ and $x_{14}^1$. Hence, another sampling subgraph $G_{sub}^0$ is constructed, and both of these two subgraphs $G_{sub}^0$ and $G_{sub}^1$ are formed into the sampling subgraph set. The above steps are performed until the sampling depth decreases to 0.

### 3.3.2. Aggregating Neighbor Features

After sampling, it is necessary to aggregate information from neighbors and update the features of the target node, in which the forward propagation direction is opposite to the sampling direction. The forward propagation is implemented by the message passing function and aggregation update function, thus extracting stroke features. These functions are defined as follows:

$$m_u^k = \varnothing\left(x_u^{k-1}, x_v^k\right), (u, v \in V) \tag{2}$$

$$h_v^k = \varphi\left(max\left\{ReLU\left(W_{pooling}m_u^k + b\right)\right\}\right) \tag{3}$$

where $\varnothing$ is the message passing function representing the transmitted characteristics of stroke defined on edge between the start node $x_u^{k-1}$ and the target node $x_v^k$. And $\varphi$ is the aggregation update function to gather the features of adjacent nodes, which usually consists of GCN aggregation, mean aggregation, and pooling aggregation. By comparing the experiment, the pooling aggregation uses the maximum pooling operator to highlight the influence degree of neighbor nodes. Therefore, the pooling aggregation is adopted to aggregate neighbor nodes and update the features of the target stroke node.

In aggregation, the set of sampling subgraphs ($\left\{ G_{sub}^k \right\}$) and the set of stroke node features ($\left\{ h_v^k, v \in V \right\}$) are taken as input, while the characteristics of the target stroke node are taken as output. The aggregation traverses in the opposite direction of sampling and uses $h_v^k$ to represent the characteristics of the node $x_v^k$ in layer $k$. Firstly, the target node $x_v^k$ is used to obtain the outer neighbor nodes $x_u^{k-1}$ and a set of their characteristics $\left\{ h_u^{k-1}, u \in N(v) \right\}$. Then, the passing information of edges between nodes is obtained to aggregate and update the characteristics with $h_{N(v)}^k$. In detail, $N(v)$ represents all of the neighbor nodes that are not aggregated, and $h_u^{k-1}$ represents node characteristics generated by the last aggregation. In this way, GNN could not only aggregate the features of neighbor strokes but also consider the topologic relationship of road networks, thus realizing deep learning in road network selection. As shown in Figure 7, the $h_{15}^2$ characteristics in layer 2 need to obtain the characteristics of $h_{11}^1$, $h_{12}^1$, and $h_{14}^1$ in layer 1 and $h_3^0$, $h_4^0$, and $h_5^0$ in layer 0. At the same time, sampling with replacement is adopted to aggregate the characteristics of $h_6^0$, $h_8^0$, and $h_9^0$. The pseudo-codes of random sampling and aggregating from neighbors are as follows (Algorithm 1).

---

**Algorithm 1:** Random sampling and aggregating from neighbors for GNN.

---

**Input:** Dual graph of road network $G(V, E)$, Characteristics of stroke nodes $\{x_v, \forall v \in V\}$,
         Sampling depth of nodes $K$, Sampling function of neighbor nodes $N_k(v)$,
         Message passing function $\Phi$, Aggregation update function $\varphi$,
**Output:** The embedding characteristics of road nodes $\{h_v, \forall v \in V\}$,

  1           Take the target node as the initial subgraph $G_{sub}^k$,

  2           **for** $k \leftarrow K \cdots 1$ **do**

  3             **for** $v \in G_{sub}^k$ **do**

  4               $G_{sub}^{k-1} \leftarrow G_{sub}^{k-1} \cup N_k(v)$;

  5             **end**

  6           **end**

  7           $h_v^0 \leftarrow x_v$, $\forall v \in G_{sub}^0$;

  8           **for** $k \leftarrow 1 \cdots K$ **do**

  9             $h_v^k \leftarrow x_v$, $\forall v \in G_{sub}^k$;

10             **for** $v \in G_{sub}^k$ **do**

11               $m_u^k \leftarrow \Phi\left( h_u^{k-1}, h_v^k \right)$, $\forall u \in N_k(v)$;

12               $h_v^k \leftarrow \varphi\left( max\left\{ ReLU\left( W_{pooling} m_u^k + b \right) \right\} \right)$;

13             **end**

14           **end**

15           $h_v \leftarrow h_v^k$, $\forall v \in G_{sub}^k$;

16           **return** $h_v$;

---

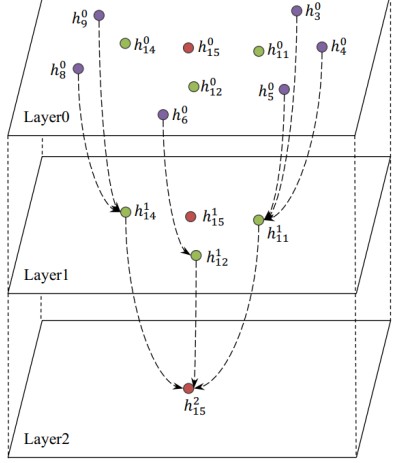

**Figure 7.** Aggregating features from neighbor stroke nodes.

*3.4. Selection Model Design*

In general, two different deep learning methods exist, that is, transductive and inductive learning [40]. The transductive learning method refers to training models with both training and test datasets. However, the inductive learning method refers to learning rules from training data and then applying rules to test data. This paper adopts Graph-SAGE, an inductive learning framework, for selection training, in which the GNN model is constructed by the space method. The defined GNN-based road network selection model includes three parts: multiple hidden layers (GraphSAGE), a classification layer, and a normalization layer. Additionally, cross-entropy loss and adaptive moment estimation methods are employed to optimize parameters, thus realizing intelligent road network selection.

As described previously, the embedding feature representation of the target node is sampled and aggregated from neighbors. As shown in Figure 8, GraphSAGE implements representation learning and computes the embedding features of the road network, while a classification layer implements task learning and computes the classification results. These two processes are iterated repeatedly to obtain the optimal feature representation and selection result. In detail, the sampling subgraph and features of the target node are taken as the input to aggregate the features from neighbors and acquire the embedding representation of strokes through multiple hidden layers. Afterward, a classification layer realizes linear transformation for stroke features. Lastly, the normalized index obtains the selection probability of stroke, thus completing road network selection.

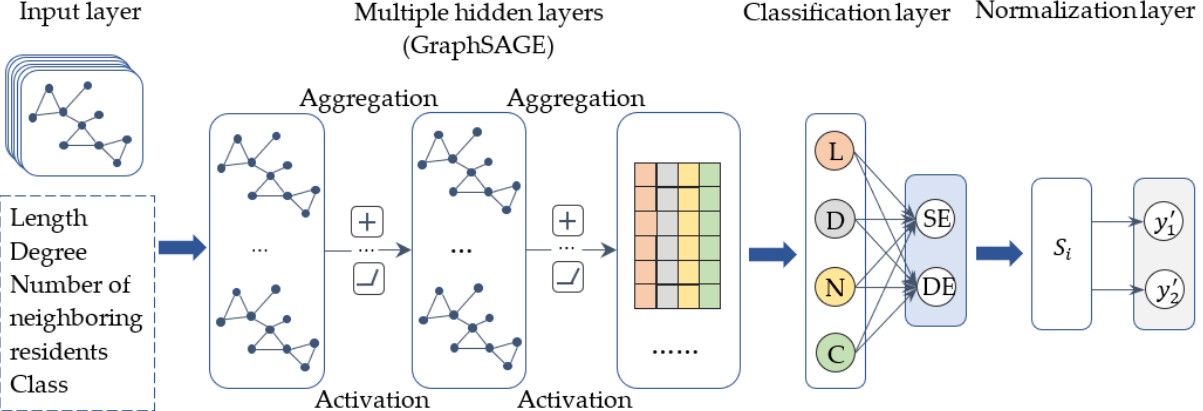

**Figure 8.** GNN-based model for road network selection.

3.4.1. Multiple Hidden Layers

Multiple hidden layers, primarily composed of GraphSAGE, are mainly used to realize the embedding representation of stroke features. In this paper, we take sampling subgraph sets as the input while taking selection (SE) and deletion (DE) as the labels of nodes. In addition, four features are taken as the attributes of nodes, such as length (L), degree (D), number of neighboring residents (N), and class (C). After inputting the sampling subgraph, the whole process is propagated through multiple hidden layers. The calculation of transmission from layer $k$ to layer $k + 1$ is as follows:

$$H^{k+1} = \sigma\left(H^k W^k + b^k\right) \tag{4}$$

where $H^k$ represents the characteristic set of nodes in the exterior layer, and $H^{k+1}$ represents the characteristic set of nodes aggregated by the exterior layer. $W^k$ and $b^k$ are the trainable weights of GraphSAGE, and $\sigma$ is the activation function. Additionally, $k$ represents the depth of sampling. From multiple hidden layers, attributes are aggregated into high-dimensional vectors to show the features of stroke nodes in the dual graph. As stated earlier in this paper, eight is the length of initial stroke features. Through aggregation by

multiple hidden layers, embedding vectors are obtained to demonstrate the importance of stroke features in different dimensions.

### 3.4.2. Classification Layer

After the aggregation from multiple hidden layers, a fully connected neural network is applied to calculate each stroke's selection or deletion values. The four-length embedding vector of the target node is used as input, which is linearly aggregated through the fully connected network. Therefore, the GNN-based model could calculate the importance of selecting and deleting to judge the selection result of each road stroke.

### 3.4.3. Normalization Layer

The normalization layer uses the softmax function to obtain selection result probability. The softmax normalized exponential function could map the multidimensional vector of any real number to another float vector with the same dimension between 0 and 1. The specific formula is as follows:

$$S_i = \frac{e^i}{\sum_{i=1}^{j} e^i} \tag{5}$$

where $S_i$ represents the output characteristic value of node $i$. $j$ is the number of output nodes, which are the selection node and deletion node. The denominator of this formula represents the sum of stroke selection and deletion values. The GNN-based model transforms selection results to the interval range [0,1] through the softmax function, representing road network selection results in probability.

### 3.4.4. GNN-Based Model Optimizing

The GNN-based model calculates road network features through forward propagation, which may output the selection results with errors in the initial stage. In forward propagation, the output layer error is calculated based on the cross-entropy loss function, and the error is returned by backpropagation. Meanwhile, the gradient descent method is adopted to optimize model parameters.

As a metric measuring the level of chaos, entropy represents the total information in the system. The lower the entropy, the smaller the error of the predicted value is for the probability of selection and deletion [41,42]. Therefore, the cross-entropy loss function is adopted in model training to measure the difference between probability and real labels. The specific formula is as follows:

$$loss = -\frac{1}{N} \sum_{v \in V} \left( y_{v1} log y'_{v1} + y_{v2} log y'_{v2} \right) \tag{6}$$

where $N$ represents the number of samples, and $V$ represents the set of target nodes. In addition, $y_{v1}$ and $y_{v2}$ are real selection results, where 1 denotes selection, and 0 denotes deletion. $y'_{v1}$ and $y'_{v2}$ represent selection and deletion calculated by the forward propagation of our model, respectively.

In addition to the cross-entropy loss function, the adaptive moment estimation (Adam) is chosen as the stochastic gradient descent optimization algorithm. The Adam method considers the change in gradient for the current and previous moment, thus smoothing parameters and reducing the gradient swing scope in the backpropagation process. At the same time, it also uses a weighted average to consider the cumulative effect of the gradient for all moments before, thus improving the convergence rate of the GNN model.

## 4. Experiments and Results

The proposed model is implemented in Pytorch using Python and uses road networks and resident data of different scales to conduct experiments. A five-layer neural network model is used to conduct experiments, including two GNN hidden layers, two fully connected neural network layers, and one normalized exponent layer. In addition, the

data from the national basic scale topographical map series at 1:10,000 scale and 1:50,000 scale are, respectively, chosen as experimental data and reference results, which consist of the road network and neighboring residents. As shown in Figure 9, we take the red-framed area as the experimental area and the rest as the case area to analyze the proposed method deeply.

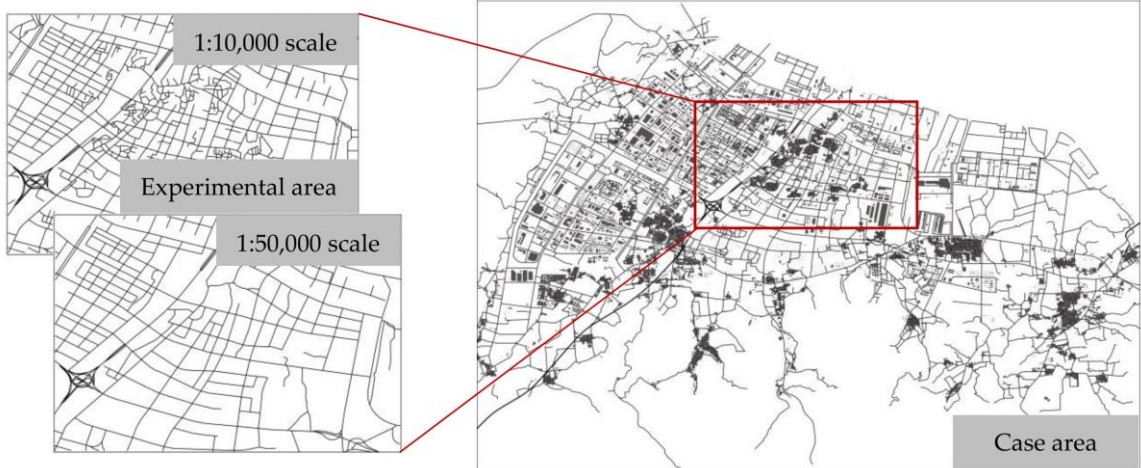

**Figure 9.** Experimental area and case area.

*4.1. Different Aggregations Analysis*

After processing by link breaking for the whole regional road network, road stroke and dual graph are constructed to calculate features and labels, thus obtaining samples. As shown in Table 2, the sample library, built from cases, contains 1685 nodes and 2961 edges. The dual graph in the case area is taken as the input to train the GNN-based model, thus intelligently selecting the road network in the experimental area.

**Table 2.** Sample library of road network selection.

| Road ID | Length | Degree | Resident Number | Class | Label |
|---------|--------|--------|-----------------|-------|-------|
| 1 | 0.0026 | 0.0349 | 0.3333 | 0,0,0,0,1 | 0,1 |
| 2 | 0.0056 | 0.0233 | 0.3333 | 0,0,0,0,1 | 0,1 |
| 3 | 0.0549 | 0.0465 | 0.6667 | 0,0,0,0,1 | 1,0 |
| ... | ... | ... | ... | ... | ... |
| 1313 | 0.2005 | 0.1163 | 0 | 0,0,0,1,0 | 1,0 |
| 1314 | 0.0363 | 0.0465 | 0.3333 | 0,0,0,1,0 | 0,1 |
| ... | ... | ... | ... | ... | ... |
| 1684 | 0.7386 | 0.3372 | 0 | 0,0,1,0,0 | 1,0 |
| 1685 | 0.5214 | 0.2907 | 0.3333 | 0,0,1,0,0 | 1,0 |

In the process of parameter adjustment, different aggregation methods would affect the training result. This paper uses GCN aggregation, mean aggregation, and pooling aggregation methods to conduct comparative experiments. During the model learning process, some parts of training data are reserved as validation data to adjust the parameters and verify generalization ability. Figure 10 illustrates the average accuracy of models with different aggregation functions, where the blue and red curves express the average accuracy of training and validation data. After 20 epochs of iterative learning, the average accuracy of the model reaches a relatively stable level and is no longer improved obviously. At the same time, it is found that the Pooling aggregation function has the best generalization effect. The pooling aggregation uses the maximum pooling operator to highlight the influence degree of neighbor nodes, which is insensitive to the sequence of nodes and effectively captures different aspects of the neighborhood set. Hence, the maximum pooling operator is adopted to aggregate the neighbor features, which could focus on the influence of essential neighbor nodes around the target node and achieve high accuracy.

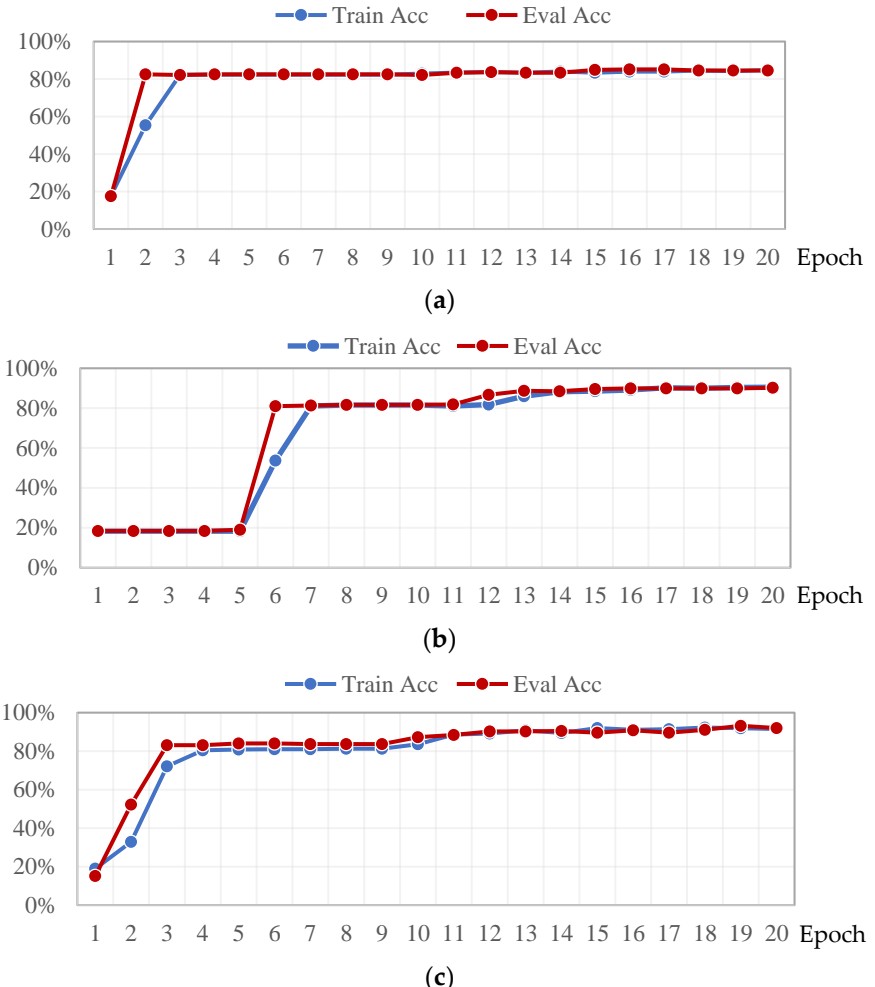

**Figure 10.** The average accuracy of different aggregation functions. (**a–c**) demonstrate the average accuracy of GCN aggregation, mean aggregation, and pooling aggregation, respectively.

### 4.2. Different Deep Learning Models Analysis

To verify the effectiveness of GNN hidden layers for aggregating stroke features, a two-layer fully connected neural network and one normalized exponent layer model without two GNN hidden layers are used for a comparative experiment. In the fully connected neural network layers, the first layer's input and output feature dimensions are defined as eight and four, and the second layer is configured as four and two. As shown in Table 3, there is the accuracy of different models for validation data. By comparison, it is found that the accuracy of the GNN model is stable at about 90%, which is higher than the comparative model. Therefore, the proposed method could not only learn the semantic, geometric, and topological features of road networks simultaneously but also learn the features of neighbor nodes, thus improving the accuracy of the selection result.

The road network of 1:10,000 scale in the experimental area is chosen to conduct a comparative experiment, thus verifying the scientificity of the GNN model. The selection result of our method is shown in Figure 11a, and the 1:50,000 scale reference result is shown in Figure 11b. The accuracy of our selection result is 91.97% by matching the entity of the selected result with the reference result. As shown in Figure 11, the red circles line out the inconsistent selection, while the blue circle lines out the inconsistent deletion. Although there are still a few inconsistencies, it is found that the result is consistent with the reference result in general.

**Table 3.** The comparison experiment of different models.

| Number of Epochs | GNN Model | Comparative Model |
| --- | --- | --- |
| 1 | 88.43% | 81.25% |
| 2 | 90.21% | 85.16% |
| 3 | 90.21% | 84.38% |
| 4 | 90.50% | 85.16% |
| 5 | 89.61% | 86.72% |
| 6 | 90.80% | 83.59% |
| 7 | 89.61% | 81.25% |
| 8 | 91.10% | 82.03% |
| 9 | 93.18% | 85.81% |
| 10 | 91.99% | 85.94% |

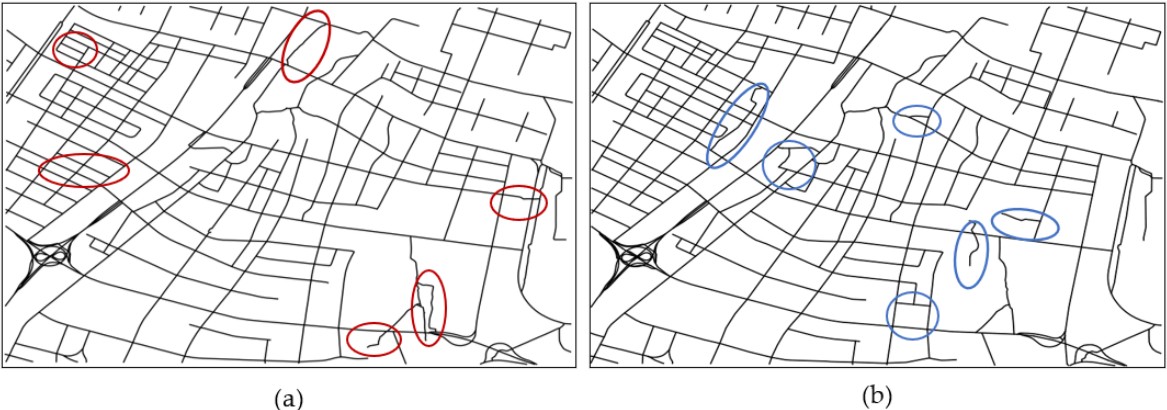

(a)          (b)

**Figure 11.** The comparative analysis demonstrating (**a**) selection result and (**b**) reference result on a 1:50,000 scale.

*4.3. Comparative Experiments with Traditional AHP Method*

In order to verify the effectiveness of the whole model, we implement the traditional multiple criteria decision-making method based on the analytic hierarchy process (AHP) [19] for comparison. The AHP method allows the relative prioritization and assessment of road strokes under multiple features. We take class, length, degree, and resident number as features, and thus it is the same with this paper. As presented in the two enlarged portions of the results in Figure 12, the traditional AHP method eliminates strokes with high connectivity, while our method could retain them circled by red lines. Therefore, the advantage of our method is that it can consider aspects of both attribute and spatial structure.

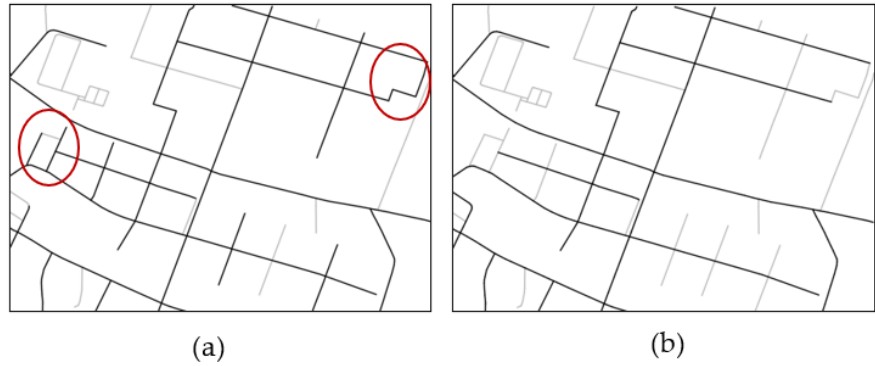

(a)          (b)

**Figure 12.** The enlarged portions of the results demonstrating (**a**) generalization by GNN model and (**b**) generalization by AHP method.

In this study, the results of the selection are also evaluated from quantitative assessment inspection by recall rate (*R*), precision rate (*P*), $F_1$-score, and Matthews correlation coefficient (MCC). The measurements could be denoted as

$$R = \frac{TP}{TP + FN} \tag{7}$$

$$P = \frac{TP}{TP + FP} \tag{8}$$

$$F_1\text{-score} = \frac{2 \times R \times P}{R + P} \tag{9}$$

$$MCC = \frac{TP \times TN - FP \times FN}{\sqrt{(TP + FP) \times (TP + FN) \times (TN + FP) \times (TN + FN)}} \tag{10}$$

where *TP* represents both the positive true value and positive predicted value and is calculated by the coincident length of the selection result and reference result; *FP* represents the negative true value and positive predicted value and is calculated by the selection length minus coincident length; *FN* represents the positive true value and negative predicted value, and is calculated by the reference length minus coincident length; *TN* represents both of the negative true value and predicted value, and is calculated by the coincident length of detection result and reference result. Table 4 shows the quantitative assessment results of different selection methods, which represent all the higher values of the GNN-based model than those of the AHP-based model. Therefore, our method is effective for road network selection.

**Table 4.** The quantitative assessment results of different models.

| Selection Models | Recall Rate (*R*) | Precision Rate (*P*) | $F_1$-Score | MCC |
|---|---|---|---|---|
| GNN-based model | 92.21% | 91.97% | 0.921 | 0.664 |
| AHP-based model | 88.67% | 87.42% | 0.880 | 0.500 |

*4.4. Comparative Experiments with Generalization Tool in ArcGIS*

In order to further verify the rationality of the proposed method, the generalization tool in ArcGIS is adopted to conduct another comparative experiment. The tool generates a simplified road network that retains connectivity at a smaller scale. The selection result is determined by feature significance, importance, and density. In detail, the density is determined by the minimum length parameter, which corresponds to the shortest segment. According to the current experimental results, the minimum length parameter is set as 10 m to realize the selection in ArcGIS.

If the overall selection proportion is guaranteed to be fixed, the significant roads selected by the above two methods are the same, such as national, provincial, and county highways. However, the GNN-based method could retain more urban main roads. Moreover, the ArcGIS method could keep 196 complete meshes in terms of structure, while the GNN-based could maintain 200 meshes. Regarding the hanging stroke, there are 18 road sections in the ArcGIS selection result and 15 in the GNN-based selection result. As shown in Figure 13, the ArcGIS tool generates more suspended roads, which are marked by dotted lines, and our method could retain more complete strokes, thus maintaining the overall structure of the road network better.

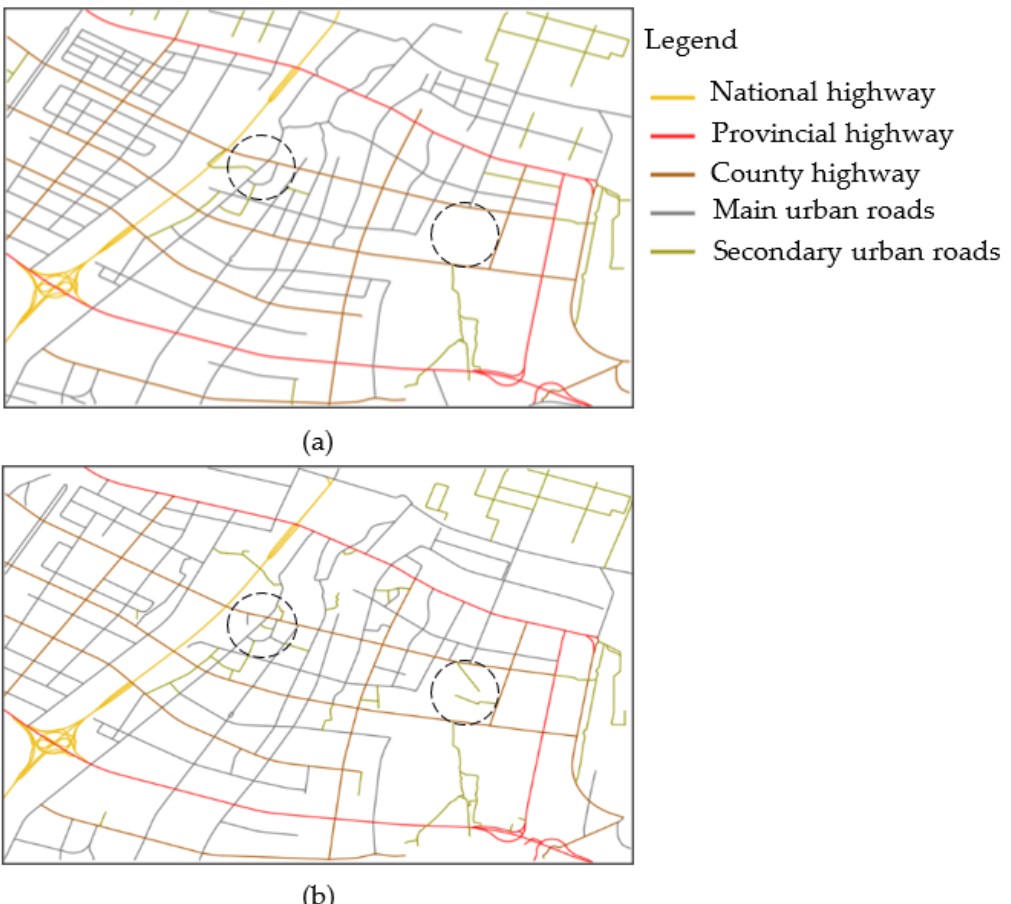

**Figure 13.** The road selection results demonstrating (**a**) generalization by GNN model and (**b**) generalization by the generalization tool in ArcGIS.

## 5. Conclusions

As a hotspot of deep learning research, GNN could simultaneously learn the attribute characteristics and spatial structure of road networks. Therefore, this paper proposes a deep learning framework for selection, which contains a sampling method and a GNN-based model to complete the classification task automatically. Firstly, the selection case is designed, thus constructing a sample library. Secondly, neighbor sampling and aggregation methods are developed to update the attributes based on the spatial structure of stroke nodes. Then, this paper designs the selection model with three parts, including two GNN hidden layers, two fully connected neural network classification layers, and a normalization layer. At the same time, the cross-entropy loss function and adaptive moment estimation methods are used to optimize the model parameters. Finally, the proposed method is verified through comparative analysis.

Road network selection is one of the major concerns in cartographic generalization and has extensive applications such as urban planning and navigation. Compared with previous studies, this paper could automatically learn the road network's semantic, geometric, and topological features from the sample library. The selection result of our method is most similar to the reference result, thus providing a new idea for applying deep learning in road selection and facilitating more intelligent cartographic generalization methods.

However, there are some limitations to our approach. As mentioned in Section 3.2.1, the method only takes into account four selection features. In the future, we plan to consider multiple factors, such as hilly areas contours, traffic flow information, and POI distribution. On the other hand, we plan to use much more variants of GNN to improve this work so that such a framework can also be applied to different spatial scales and other map features.

**Author Contributions:** Conceptualization, Xuan Guo; methodology, Xuan Guo and Junnan Liu; software, Junnan Liu; resources, Fang Wu and Haizhong Qian; writing—original draft preparation, Xuan Guo and Junnan Liu; writing—review and editing, Haizhong Qian and Fang Wu. All authors have read and agreed to the published version of the manuscript.

**Funding:** This work was supported by the Excellent Youth Foundation of Henan Scientific Committee, grant number 212300410014, the National Natural Science Foundation of China, grant number 42271463, State Key Laboratory of Geo-Information Engineering, grant number SKLGIE2023-M-4-1, and the Key Research and Development Project of Henan Province (Science and Technology), grant number 232102211026.

**Data Availability Statement:** The current study's data are available from the corresponding author based on a reasonable request.

**Acknowledgments:** We would like to thank the anonymous reviewers for their insightful comments and substantial help in improving this article.

**Conflicts of Interest:** The authors declare no conflict of interest.

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
