# Peer review of "A Method for Intelligent Road Network Selection Based on Graph Neural Network"

_ijgi, doi:10.3390/ijgi12080336_

Round 1
Reviewer 1 Report (Previous Reviewer 3)
Authors have addressed most of the comments from the first round of review.
Author Response
Thanks for the reviewer’s comments.

Reviewer 2 Report (New Reviewer)
Overall the article is well-written. Following points must be addressed to improve its quality
1) Matrices used are recall rate (?), accuracy rate (?), and ?1 -score. The authors may include more matrices such as mathews coefficient and/or Markedness (MK) matrix.
2) The literature review must be strenghted by including some latest studies regarding the use of AI in Geosciences and Engineering such as https://www.tandfonline.com/doi/full/10.1080/10298436.2021.1904237 and https://www.sciencedirect.com/science/article/pii/S0098300423000687
3) Correct the following: "The transductive learning method refers to training models with 354 training and test data sets, which are verified further by test data". Either use the word independent test data or validation dataset
4) State the time taken by the algorithm to complete the learning.
5) The abstract is too wordy, reduce it and also present some quantative results.
6) Discussions and Conclusions section is too brief. I suggest just say Conclusions.
Minor editing of English language required
Author Response
Thanks for the reviewer’s comments. The responses to comments are provided in a point-to-point way. The manuscript has been substantially checked, revised, and polished. Please see the attachment.

Reviewer 3 Report (New Reviewer)
The manuscript is an interesting scientific investigation related to cartographic generalization. The author(s) have proposed and applied road network selection method based on graph neural network (GNN). They assert, their devised method facilitate a more intelligent road network selection method and thus could bridge the gap between deep learning and cartographic generalization. However, the authors are very selective in applying their proposed solution and have ignored fundamental principles of Cartography/ cartographic generalization.
1. Introduction
1.1. Good start of introduction is acknowledged. It can be improved and made better by adding one/two paragraphs about why cartographic generalization is necessary? And why it is subjective as well as scale related.
1.2. Line 38-39: …”Road network selection is the first and most crucial step, reducing road elements by deleting relatively unimportant parts”. What is the definition of relative unimportant? How it is decide? Please elaborate.
1.3. Line 44-45: “However, stroke features cannot reflect all aspects of network structure, thus posing several challenges to the road network selection”. It would be better to include some more challenges as only two challenges have been mentioned.
1.4. Line 57. The mentioned motivational factors i.e. evaluates semantic, geometric, and topological features need explanation especially the semantic aspect that is not explained.
1.5. Line 59: The statement, “improved selection method for generalizing road network based on GNN is proposed” needs justification in detail as it is the crux/worth of this article.
1.6. It is suggested to add a table showing comparison of current and previous approaches.
2. Related work
2.1. Line 82: Better to say, “Cartographic generalization has always drawn attention…..”
2.2. Line 120-121: The statement, “ However, none of them pay attention to variable relationships between strokes, and the characteristics could only be evaluated in one dimension” needs elaboration for better understanding of readers as they all may not be necessarily cartographers.
2.3. Line 170: It seems contradictory to statement given under abstract. “This paper adopts GNN for road network selection to learn graph structure features, which could simultaneously explore semantic, geometric, and topological features”.
3. Methods
3.1. Table 1. Better to introduce terms “class, and degree” before inserting the table.
3.2. Figure 3.
· Roads are described by polygons in databases for large scale mapping (1:10,000) and in smaller scale (1:50,000) roads are defined by lines. But it is not mentioned.
· If geometry of both roads is different then Figure 3 is not valid. Please explain/rectify.
· Also roads are represented as symbols on scale 1;50,000 that are no longer true to scale. This point needs attention and to be discussed.
· Human interaction remains necessary during cartographic generalization especially in conceptual (semantic) generalization. But it is not mentioned in Figure 3.
· After generalization, positional accuracy of spatial objects will reduce but how much?
3.3. Please define basic rules e.g. aggregation (merging), emphasis (enhancement) and exaggeration (enlargement) and classification before Figure 4.
3.4. Similarly, priority rules are missing. For example regarding displacement of buildings/houses along the roads, water features, contours etc.
3.5. Relationship between features should be taken into account. In case of roads in hilly areas, after applying the proposed solution, the contours will still fit to the road network? Please elaborate.
3.6. It is suggested to apply the proposed solution on map/database having multiple layers such as hydrology, contours, and buildings in addition to road network.
My overall recommendation is Major Revision in the manuscript.

Moderate editing is essentially required
Author Response
Thanks for the reviewer’s comments. The responses to comments are provided in a point-to-point way. The manuscript has been substantially checked, revised, and polished. Please see the attachment.

Round 2
Reviewer 2 Report (New Reviewer)
The authors have done a considerable job in revising the manuscript which can now be accepted.
minor issues/editing
Reviewer 3 Report (New Reviewer)
The authors have addressed my concerns in the revised manuscript.
This manuscript is a resubmission of an earlier submission. The following is a list of the peer review reports and author responses from that submission.
Round 1
Reviewer 1 Report
This paper proposes a road network selection method developed mainly based on multiple GNN layers, which aims to learn the attribute and spatial structure of road network at the same time. The content of manuscript is self-contained and is technically correct. The main problem existed in this manuscript, as far as I’m concerned, is the lack of quantitative analysis to verify the effectiveness of the proposed method. Also, the comparative experiments somewhat let me down.
1. The test data as shown in Fig.10 is better to be described detailly. Like what the type and size of those roads? Where and how are the referenced road obtained? And so on.
2. 4.1 indicates that the pooling aggregation function performs best and it is thus used in the following processing. This conclusion is not convincing as the three aggregation algorithms are tested on only one image. If testing on more road images, other two functions are likely to perform better.
3. Using a 2-layer neural network model as a comparison is not fair. Neural network model with more layers theoretically attains better result in most of cases. Why not use 3- or 4-layer neural network models? and, why not use other widely used GNN rather than NN as the comparison? This seems more reasonable.
4. The merit of the proposed method can cope with both attribute and spatial structure of road network. However, I didn’t see any tests on them separately and this merit is not validated well. For example, you can use only attribute information or only spatial structure information to obtain results in the comparison.
5. I’m not sure the reason that made the comparison with Arcgis. The statement in 4.3 is likely irrelevant to Fig.13. I suggest to add one more representative method in comparisons, such as those method mentioned in Introduction, rather than only deep learning-based methods.
6. More tested images and quantitatively analysis are strongly encouraged.
Based on abovementioned issues, I recommend to a major revision.
Reviewer 2 Report
The paper proposed a road network selection method based on deep learning method. It seems good through the comparative experiments. To improve this paper, I have some suggestions.
1. Does the proposed method are applicable to different spatial scales of images?
2. Please explain the innovation point of the proposed method more clearly in both sections of Introduction and Conclusion.
3. I suggest the authors to add a discussion section to describe the advantages and disadvantages of the proposed approach with traditional and state-of-the-art methods.
Reviewer 3 Report
The manuscript proposes a Graph Neural Network oriented solution of road network selection. The topic is significant and within the scope of journal. The manuscript is well written and ticks all the essential boxes. Followings are a few of my minor comments to further enhance the quality and readability:
1. Please ensure that all the keywords are mentioned within the Abstract.
2. After line 67, It would be better for readability to put the contributions as a list rather than paragraphs.
3. Under section 3.1., please crosscheck the list format. It should be standard list. This applies to all the lists in the manuscript.
4. The implications of study are missing. An example use-case from real world where the proposed solution can be deployed will be very helpful in the discussions section.
5. The limitations of the work are missing in the Conclusion and so are the future work directions. It is advised to clearly highlight the limitations and propose the future directions for the presented research.
6. A fundamental question on why authors decided to choose Graph Neural Networks in the first place? What was the intuition behind the selection? It should be clearly justified in the Introduction section.